# UV-Spectrophotometric Determination of the Active Pharmaceutical Ingredients Meloxicam and Nimesulide in Cleaning Validation Samples with Sodium Carbonate

Pavel Anatolyevich Nikolaychuk 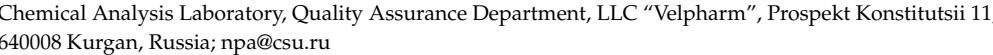

Chemical Analysis Laboratory, Quality Assurance Department, LLC "Velpharm", Prospekt Konstitutsii 11, 640008 Kurgan, Russia; npa@csu.ru

**Abstract:** The spectrophotometric methods of determination of the active pharmaceutical ingredients meloxicam and nimesulide were reviewed and a simple UV-spectrophotometric method for the determination of these active pharmaceutical ingredients in industrial equipment cleaning validation samples was proposed. The methods were based on extraction of the residual quantities of meloxicam and nimesulide from the manufacturing equipment surface by the concentrated sodium carbonate solution and the subsequent UV-spectrophotometric determination of the basic forms of the drugs at the wavelength of 362 nm for meloxicam and at 397 nm for nimesulide. The calibration graphs were linear in the range from 5 to 25 mg/L of both nimesulide and meloxicam, the molar attenuation coefficients were 6100 $m^2$/mol for nimesulide and 9100 $m^2$/mol for meloxicam, the limit of detection was 0.8 mg/L for nimesulide and 1.9 mg/L for meloxicam and the limit of quantification was 2.5 mg/L for nimesulide and 5.8 mg/L for meloxicam. The methods were selective with respect to the common excipients, showed a good accuracy (the relative uncertainty did not exceed 7%) and precision (the relative standard deviation did not exceed 4%), did not require lengthy sample preparation or sophisticated laboratory equipment and were suitable for the routine analysis of cleaning validation samples.

**Keywords:** meloxicam; nimesulide; UV-spectrophotometric determination; cleaning validation samples



## 1. Introduction

Meloxicam (IUPAC name: 4-Hydroxy-2-methyl-N-(5-methyl-2-thiazolyl)-2H-1,2-benzothiazine-3-carboxamide-1,1-dioxide, CAS number: 71125-38-7) and nimesulide (IUPAC name: N-(4-Nitro-2-phenoxyphenyl)methanesulfonamide, CAS number: 51803-78-2) are both widely used nonsteroidal anti-inflammatory drugs. Meloxicam was developed for the treatment of rheumatoid arthritis and osteoarthritis [1,2]; nimesulide was found to be effective in reducing pain associated with osteoarthritis, cancer, thrombophlebitis, oral surgery and dysmenorrhea [3,4].

When several pharmaceuticals are manufactured on the same production line, the pharmaceutical product can be contaminated by other pharmaceutical products, by cleaning agents, by microorganisms or by other materials. The procedure of cleaning the industrial equipment apparatus as well as the processing area is required to effectively remove the potentially dangerous substances from it. However, it is necessary to validate the cleaning procedures to ensure the safety, efficacy and quality of the subsequent batches of drug product [5]. Historically, the cleanliness of equipment manufacturing is validated and verified using direct swabbing of the equipment and subsequent analytical testing of the swab extracts [6]. The quantitative determination of meloxicam and nimesulide is possible using a variety of methods, including all types of chromatographic, spectroscopic and voltammetric techniques [7]. A routine determination of the pharmaceutical ingredients in the swab extracts however should ideally be performed directly in the production area, should not require comprehensive equipment and the method should be rapid and simple. Therefore, the method utilizing UV-visible spectroscopy is preferred. The existing

spectrophotometric methods for the determination of nimesulide [8–32] are summarized in Table 1 and those for meloxicam [32–63] in Table 2.

**Table 1.** A review of spectrophotometric methods of determination of nimesulide.

| Solvent | Used Reagents | Wavelength, nm | Linearity, mg/L | Accuracy, % | Precision, % | Reference |
|---|---|---|---|---|---|---|
| Methanol | None | 397 | Not specified | Not specified | Not specified | [8] |
| Water | NaOH | 397 | 5–30 | 2 | 1 | [9] |
| Water | Phosphate buffer | 393 | Not specified | Not specified | Not specified | [10] |
| Methanol | Iminodibenzyl | 600 | 0.1–7.5 | 0.2 | 0.1 | [11] |
| Methanol | 3-aminophenol | 470 | 0.4–12 | 0.3 | 0.2 | [11] |
| Ethanol | None | 262–291 Second derivative | 2–90 | 2 | 1 | [12] |
| Chloroform | None | 248–268 Second derivative | 2–50 | 3 | 1 | [12] |
| Water/chloroform | Hexadecyl-trimethyl-ammonium bromide | 404 | 6–20 | 1 | 0.7 | [13] |
| Water/chloroform | Bromocresol green | 412 | 2–18 | 5 | 0.6 | [14] |
| Water/chloroform | Bromocresol purple | 410 | 2–16 | 4 | 0.5 | [14] |
| Water/chloroform | Bromothymol blue | 407 | 2–18 | 3 | 0.5 | [14] |
| Water/chloroform | Brilliant blue G | 502 | 2–18 | 5 | 0.5 | [14] |
| Water/chloroform | Methyl orange | 482 | 2–14 | 3 | 0.7 | [14] |
| Water | p-N,N-dimethyl phenylene diamine dihydrochloride, chloramine-T | 540 | 10–50 | 0.8 | 0.6 | [15] |
| Water | p-N,N-dimethyl phenylene diamine dihydrochloride, 3-methyl-2-benzothiazolinone hydrazine hydrochloride | 600 | 12.5–75 | 1.2 | 0.4 | [15] |
| Water | $HNO_2$, cresyl fast violet acetate | 565 | 2–12 | 2.2 | 0.2 | [15] |
| Water | p-methyl aminophenol sulphate, $K_2Cr_2O_7$ | 550 | 20–120 | 0.8 | 0.5 | [15] |
| Water | Thymol | 476 | 5–40 | 2.4 | 2.2 | [16] |
| Water | NaOH | 397 | Not specified | Not specified | Not specified | [17] |
| Water | NaOH | 397 | Not specified | Not specified | Not specified | [18] |
| Water/acetonitrile | None | 300 | 10–50 | 1 | 0.4 | [19] |
| Acetonitrile | None | 300 | 10–50 | 0.4 | 0.4 | [19] |
| Methanol | Orcinol | 465 | 0.4–4 | 1.8 | 1.6 | [20] |
| Water | NaOH | 460 | 0.4–5.1 | 8 | Not specified | [21] |
| Methanol/water | Phloroglucinol, ammonium sulfamate | 400 | 4–20 | 2 | Not specified | [22] |
| Methanol/water | p-dimethylamino benzaldehyde | 415 | 4–24 | 2 | Not specified | [22] |

**Table 1.** *Cont.*

| Solvent | Used Reagents | Wavelength, nm | Linearity, mg/L | Accuracy, % | Precision, % | Reference |
|---|---|---|---|---|---|---|
| Methanol | $CuSO_4$, $KNaC_4H_4O_6$, KI, NaOH | 400 | 25–200 | 0.8 | 2.1 | [23] |
| Ethanol/water | Bromocresol green | 643 | 2–14 | 0.5 | 1.2 | [24] |
| Ethanol/water | Bromocresol purple | 437 | 2–12 | 0.5 | 1.6 | [24] |
| Ethanol/water | Brilliant blue G | 554 | 2–13 | 1 | 1.3 | [24] |
| Methanol/water | N-bromosuccinimide, promethazine hydrochloride | 610 | 0.4–8 | Not specified | Not specified | [25] |
| Methanol | None | 297 | 10–50 | 2 | Not specified | [26] |
| Methanol/acetonitrile | None | 295 | 10–50 | 2 | Not specified | [26] |
| Methanol | Folin–Ciocalteu reagent | 600 | Not specified | Not specified | Not specified | [27] |
| Water | NaOH | 393 | 1.5–14 | Not specified | Not specified | [28] |
| Methanol/water | 8-hydroxy-quinolinol | 480 | 0.5–25 | 1.6 | 1.2 | [29] |
| Water | Sodium citrate, phenol | 390 | 10–40 | 3.6 | Not specified | [30] |
| Water | Sodium benzoate, phenol | 390 | 10–50 | 1.5 | Not specified | [31] |
| Water | $KMnO_4$, Fast green FCF | 625 | Not specified | Not specified | Not specified | [32] |
| Water | $Na_2CO_3$ | 397 | 5–25 | 6.4 | 3.4 | This work |

**Table 2.** A review of spectrophotometric methods of determination of meloxicam.

| Solvent | Used Reagents | Wavelength, nm | Linearity, mg/L | Accuracy, % | Precision, % | Reference |
|---|---|---|---|---|---|---|
| Water | $KMnO_4$, Fast green FCF | 625 | Not specified | Not specified | Not specified | [32] |
| Methanol | $FeCl_3$ | 570 | 2–200 | 2.3 | Not specified | [33] |
| Water | NaOH | 362 | 0.5–20 | 1.9 | Not specified | [33] |
| Water | Phosphate buffer | 362 | Not specified | Not specified | Not specified | [34] |
| Methanol/acetonitrile | $AlCl_3$ | 375 | 5–30 | 2.7 | 1.8 | [35] |
| Ethanol | HCl, NaOH | 340–384 Difference spectrum | 2–10 | 0.5 | 0.8 | [36] |
| Ethanol | HCl | 322–368 First derivative | 1–10 | 0.5 | 1.3 | [36] |
| | HCl | 343–385 Second derivative | 1–10 | 0.5 | 0.6 | [36] |
| Water/chloroform | Saframin T | 518 | 4–12 | 1 | 0.4 | [36] |
| Water | N-bromosuccinimide, chloranilic acid | 530 | 10–160 | 8 | 1.2 | [37] |
| Water/1,4-dioxan | $UO_2(NO_3)_2$ | 398 | 5–60 | 1 | 1.5 | [38] |
| Water/ethanol | $AgNO_3$ | 412 | 1–15 | Not specified | 1.3 | [39] |

**Table 2.** *Cont.*

| Solvent | Used Reagents | Wavelength, nm | Linearity, mg/L | Accuracy, % | Precision, % | Reference |
|---|---|---|---|---|---|---|
| Methanol/water | 3-Methyl-2-benzothiazolinone-hydrazone hydrochloride, ceric ammonium sulphate | 450 | 2–20 | 1.0 | 0.5 | [40] |
| Water | NaOH | 269 | 5–30 | 0.3 | 4.2 | [41] |
| Water | FeCl$_3$ | 476 | 50–250 | 0.5 | 2 | [41] |
| Water | Trisodium citrate | 269 | 5–30 | 2.3 | 5.7 | [41] |
| Water | Sodium nitroprusside, hydroxylamine | 363 | 4–20 | 3.8 | 1.5 | [42] |
| Methanol/water | FeCl$_3$, 1,10-phen-anthroline | 343 | 10–50 | 1.5 | 0.9 | [42] |
| Water | FeCl$_3$, K$_3$[Fe(CN)$_6$] | 770 | 0.25–2.5 | 1.2 | Not specified | [43] |
| Water | Folin–Ciocalteu reagent | 740 | 5–15 | 0.4 | Not specified | [43] |
| Water/1,4-dioxan/acetonitrile | HCl | 341 | 6–14 | 2.3 | 1.8 | [44] |
| Water | Procaine benzylpenicillin | 492 | 5–80 | Not specified | Not specified | [45] |
| Water | p-methyl aminophenol sulfate, NaIO$_4$ | 656 | 15–225 | Not specified | Not specified | [45] |
| Methanol/water/chloroform | Methylene blue | 654 | 1–5 | 1.2 | 2.3 | [46] |
| Acetonitrile | 2,3-dichloro-5,6-dicyano-p-benzoquinone | 455 | 40–160 | 1 | 1 | [46] |
| Methanol/water | Borate buffer | 363 | 0.5–30 | 1 | 1.4 | [47] |
| Water | FeCl$_3$, K$_3$[Fe(CN)$_6$] | 770 | 10–25 | 5 | Not specified | [48] |
| Methanol/water | HCl | 346 | 5–150 | 3 | 0.5 | [49] |
| Water | N-bromosuccinimide, indigo carmine | 610 | 0.2–50 | 1.5 | Not specified | [50] |
| Methanol/water | NaOH | 365 | 2–12 | 1.1 | 1.3 | [51] |
| Water | Phosphate buffer | 360 | 2–12 | 1.6 | 1.1 | [51] |
| Methanol | UO$_2$CO$_3$ | 406 | 10–100 | 1 | Not specified | [52] |
| Methanol | FeCl$_3$ | 580 | 37.5–300 | 1 | Not specified | [52] |
| Ethanol | FeCl$_3$, K$_3$[Fe(CN)$_6$] | 708 | 0.1–11 | 1.3 | 0.7 | [53] |
| Water | Orange G | 358 | 1–22 | 0.4 | 0.2 | [54] |
| Water | Methylene blue | 652 | 1–22 | 0.2 | 0.2 | [54] |
| Water | CuCl$_2$ | 361 | 1–22 | 0.2 | 0.2 | [54] |
| Water/chloroform | Bromocresol green | 415 | 10–50 | 0.8 | Not specified | [55] |
| Water | NaOH | 361 | 4–14 | 1.2 | Not specified | [56] |
| Water | NaOH | 270 | 4–14 | 4.2 | Not specified | [56] |
| Water | NaOH | 215 | 4–14 | 5.5 | Not specified | [56] |

**Table 2.** *Cont.*

| Solvent | Used Reagents | Wavelength, nm | Linearity, mg/L | Accuracy, % | Precision, % | Reference |
|---------|---------------|----------------|-----------------|-------------|--------------|-----------|
| Water | NaOH | 386 First derivative | 4–14 | 1.3 | Not specified | [56] |
| Water | NaOH | 340 First derivative | 4–14 | 1.5 | Not specified | [56] |
| Water | NaOH | 273 First derivative | 4–14 | 3.4 | Not specified | [56] |
| Water | NaOH | 257 First derivative | 4–14 | 4 | Not specified | [56] |
| Water | NaOH | 409 Second derivative | 4–14 | 1.5 | Not specified | [56] |
| Water | NaOH | 359 Second derivative | 4–14 | 1.4 | Not specified | [56] |
| Water | NaOH | 316 Second derivative | 4–14 | 3.7 | Not specified | [56] |
| Water | NaOH | 278 Second derivative | 4–14 | 2.4 | Not specified | [56] |
| Water | NaOH | 269 Second derivative | 4–14 | 1.4 | Not specified | [56] |
| Water | NaOH | 251 Second derivative | 4–14 | 2.2 | Not specified | [56] |
| Water/acetone | 7-chloro-4-nitrobenz-2-oxa-1,3-diazole | 460 | 0.5–4 | 1.7 | 1.3 | [57] |
| Ethanol | None | 365 | 2–18 | 2.3 | 1.3 | [58] |
| Water | $NaNO_2$, HCl, sulfanilic acid | 365 | 1–20 | 3.5 | 2.3 | [59] |
| Water | NaOH | 269 | 5–30 | 1.6 | 1.4 | [60] |
| Water | NaOH | 253–279 Area under curve | 5–30 | 1.4 | 1.2 | [60] |
| Water | NaOH | 275 First derivative | 50–300 | 1.5 | 1.6 | [60] |
| Water | NaOH | 361 Fourth derivative | 5–35 | 0.6 | 3.4 | [61] |
| Water | NaOH | 264–277, 352–378 Area under curve | 5–35 | 0.7 | 1.8 | [61] |
| Water/methanol | 7-chloro-4-nitrobenz-2-oxa-1,3-diazole | 461 | 0.5–5 | 5 | 4 | [62] |
| Water | Folin–Ciocalteu reagent, $Na_2CO_3$ | 700 | 1.5–22.5 | 1.4 | Not specified | [63] |
| Water | $Na_2CO_3$ | 362 | 5–25 | 5.4 | 3.7 | This work |

These methods were checked for rapidness, simplicity and usage of the reagents common for pharmaceutical laboratory; it was found that the simplest methods (that allowed the determination of nimesulide and meloxicam content directly in the aqueous solutions without lengthy phase separation steps and sample or reagent preparation and that used only very common reagents available in any pharmaceutical laboratory) were based on the formation of the colored deprotonated forms of nimesulide and meloxicam in

alkaline environments. Both these active pharmaceutical ingredients exhibited an acid-base behavior and, in the presence of NaOH, formed the intensively colored yellow solutions. However, the usage of the concentrated alkalis for swabbing the drug residues from the manufacturing equipment surface was not favorable, because the alkalis themselves were toxic and could contaminate the subsequent products. The solution of sodium carbonate was much less toxic, but its usage for the determination of nimesulide and meloxicam in an aqueous solution has not yet been reported. Therefore, this study aimed to develop a method for the spectrophotometric determination of nimesulide and meloxicam in industrial equipment cleaning validation samples using sodium carbonate.

## 2. Materials and Methods

### 2.1. Reagents and Equipment

Sodium carbonate (chemically pure, 99.8%) was purchased from *Lenreaktiv*. Nimesulide (EP CRS grade), meloxicam (EP CRS grade), polyvinylpyrrolidone K-17 (USP RS grade), lactose monohydrate (reagent grade, sodium starch glycolate (reagent grade), colloidal silicon dioxide (USP RS grade), microcrystalline cellulose (reagent grade), talcum (USP RS grade) and magnesium stearate (reagent grade) were purchased from *Sigma-Aldrich*. Different tablets containing nimesulide and meloxicam were purchased from the local market. The flat plates made of stainless steel 12X12H10T were used to model the cleaning of industrial equipment. The analytical balance *Sartorius Cubis MSA 225P-ICE-DI* was used for weighting. The various micropipettes manufactured by *Thermo Fisher Scientific* were used for taking aliquots. The spectrophotometer *Mettler Toledo UV7* was used for colorimetric measurements. The chemical glassware of the 2nd grade was used. Water for preparation of solutions was twice distillated and then deionized with a *Sartorius Arium Pro VF Ultrapure Water* system.

### 2.2. Preparation of the 10% Solution of Sodium Carbonate

A total of 200.00 g of sodium carbonate was weighted and dissolved in ca. 1900 mL of water with the help of heating. The solution was cooled and transferred to the 2000 mL volumetric flask and the volume of the solution was adjusted by water.

### 2.3. Preparation of the 50 mg/L Stock Solution of Nimesulide

A total of 0.0125 g of nimesulide was weighted and dissolved in ca. 200 mL of 10% solution of sodium carbonate. The solution was transferred to the 250 mL volumetric flask and the volume of the solution was adjusted by 10% solution of sodium carbonate.

### 2.4. Preparation of Working Solutions of Nimesulide

The working solutions of nimesulide with different concentrations ranging from 5 to 25 mg/L were prepared by appropriate dilution of the stock solution with 10% solution of sodium carbonate. The working solutions were prepared daily.

### 2.5. Preparation of Sample Solutions of Nimesulide from Tablets

The tablets available on the Russian local market contained 100 mg of nimesulide. The content of ten tablets was thoroughly mixed in a porcelain mortar, collected into a beaker and dissolved in ca. 800 mL of 10% solution of sodium carbonate. The solution was transferred to the 1000 mL volumetric flask, dissolved in 10% solution of sodium carbonate and the volume of the solution was adjusted by 10% solution of sodium carbonate. Different aliquots ranging from 2.5 to 12.5 mL of the prepared solution were taken, transferred to the 500 mL volumetric flasks and the volume of the solutions was adjusted by 10% solution of sodium carbonate. The concentrations of nimesulide in the resulting solutions were equal to 5, 10, 15, 20 and 25 mg/L.

### 2.6. Preparation of Swab Extracts of Nimesulide from Working Solution

The aliquots of 10.0 mL of the prepared working solutions with different concentrations of nimesulide ranging from 5 to 25 mg/L were taken, placed onto the flat plates made of stainless steel 12X12H10T and allowed to dry in the fume hood. In the test tubes, 10.0 mL of 10% solution of sodium carbonate was prepared. The cotton swabs were dunked with 10% solution of sodium carbonate and the plates were swabbed several times during 2 min. The used swabs were immersed into the test tubes with 10% solution of sodium carbonate and mixed thoroughly for 5 min. The resulting solutions were transferred to the 10 mL volumetric flasks and the volumes of the solutions were adjusted by 10% solution of sodium carbonate. The expected concentrations of nimesulide in the swab extracts were equal to 5, 10, 15, 20 and 25 mg/L.

### 2.7. Preparation of Swab Extracts of Nimesulide from Tablets

The content of ten tablets was thoroughly mixed in a porcelain mortar, collected into a beaker and dissolved in ca. 800 mL of 10% solution of sodium carbonate. The solution was transferred to the 1000 mL volumetric flask, dissolved in 10% solution of sodium carbonate and the volume of the solution was adjusted by 10% solution of sodium carbonate. Different aliquots ranging from 2.5 to 12.5 mL of the prepared solution were taken, transferred to the 500 mL volumetric flasks and the volume of the solutions was adjusted by 10% solution of sodium carbonate. The aliquots of 10.0 mL of the prepared solutions were taken, placed onto the flat plates made of stainless steel 12X12H10T and allowed to dry in the fume hood. In the test tubes, 10.0 mL of 10% solution of sodium carbonate was prepared. The cotton swabs were dunked with 10% solution of sodium carbonate and the plates were swabbed several times for 2 min. The used swabs were immersed into the test tube with 10% solution of sodium carbonate and mixed thoroughly for 5 min. The resulting solutions were transferred to the 10 mL volumetric flasks and the volume of the solution was adjusted by 10% solution of sodium carbonate. The expected concentrations of nimesulide in the swab extract were equal to 5, 10, 15, 20 and 25 mg/L.

### 2.8. Preparation of the 50 mg/L Stock Solution of Meloxicam

A total of 0.0125 g of meloxicam was weighted and dissolved in ca. 200 mL of 10% solution of sodium carbonate. The solution was transferred to the 250 mL volumetric flask and the volume of the solution was adjusted by 10% solution of sodium carbonate.

### 2.9. Preparation of Working Solutions of Meloxicam

The working solutions of meloxicam with different concentrations ranging from 5 to 25 mg/L were prepared by appropriate dilution of the stock solution with 10% solution of sodium carbonate. The working solutions were prepared daily.

### 2.10. Preparation of Sample Solutions of Meloxicam from Tablets

The tablets available on the Russian local market contained 15 mg of meloxicam. The content of ten tablets was thoroughly mixed in a porcelain mortar, collected into a beaker and dissolved in ca. 800 mL of 10% solution of sodium carbonate. The solution was transferred to the 1000 mL volumetric flask, dissolved in 10% solution of sodium carbonate and the volume of the solution was adjusted by 10% solution of sodium carbonate. Different aliquots ranging from 16.7 to 83.3 mL of the prepared solution were taken, transferred to the 500 mL volumetric flasks and the volume of the solutions was adjusted by 10% solution of sodium carbonate. The concentrations of meloxicam in the resulting solutions were equal to 5, 10, 15, 20 and 25 mg/L.

### 2.11. Preparation of Swab Extracts of Meloxicam from Working Solution

The aliquots of 10.0 mL of the prepared working solution with different concentrations of meloxicam ranging from 5 to 25 mg/L were taken, placed onto the flat plates made of stainless steel 12X12H10T and allowed to dry in the fume hood. In the test tubes, 10.0 mL

of 10% solution of sodium carbonate were prepared. The cotton swabs were dunked with 10% solution of sodium carbonate and the plates were swabbed several times for 2 min. The used swabs were immersed into the test tubes with 10% solution of sodium carbonate and mixed thoroughly for 5 min. The resulting solutions were transferred to the 10 mL volumetric flasks and the volumes of the solutions were adjusted by 10% solution of sodium carbonate. The expected concentrations of meloxicam in the swab extract were equal to 5, 10, 15, 20 and 25 mg/L.

### 2.12. Preparation of Swab Extracts of Meloxicam from Tablets

The content of ten tablets was thoroughly mixed in a porcelain mortar, collected into a beaker and dissolved in ca. 800 mL of 10% solution of sodium carbonate. The solution was transferred to the 1000 mL volumetric flask, dissolved in 10% solution of sodium carbonate and the volume of the solution was adjusted by 10% solution of sodium carbonate. Different aliquots ranging from 16.7 to 83.3 mL of the prepared solution were taken, transferred to the 500 mL volumetric flasks and the volume of the solutions was adjusted by 10% solution of sodium carbonate. The aliquots of 10.0 mL of the prepared solutions were taken, placed onto the flat plates made of stainless steel 12X12H10T and allowed to dry in the fume hood. In the test tubes, 10.0 mL of 10% solution of sodium carbonate was prepared. The cotton swabs were dunked with 10% solution of sodium carbonate and the plates were swabbed several times for 2 min. The used swabs were immersed into the test tube with 10% solution of sodium carbonate and mixed thoroughly for 5 min. The resulting solutions were transferred to the 10 mL volumetric flasks and the volumes of the solutions were adjusted by 10% solution of sodium carbonate. The expected concentrations of meloxicam in the swab extracts were equal to 5, 10, 15, 20 and 25 mg/L.

### 2.13. General Procedure for the Determination of Nimesulide

The absorbances of the working or sample solution of nimesulide at the wavelength of 397 nm in the glass cuvette with the optical path length 1 cm were measured against the 10% solution of sodium carbonate.

### 2.14. General Procedure for the Determination of Meloxicam

The absorbances of the working or sample solution of meloxicam at the wavelength of 362 nm in the glass cuvette with the optical path length 1 cm were measured against the 10% solution of sodium carbonate.

## 3. Results
### 3.1. Selection of the Wavelength

The working solution of nimesulide with the concentration 25 mg/L and the working solution of meloxicam with the concentration 20 mg/L were prepared and their spectra against the 10% sodium carbonate solution were recorded in the quartz cuvette with the optical path length 1 cm at the wavelengths ranging from 200 to 500 nm. The spectrum of nimesulide is presented in Figure 1; it exhibits a maximum at 397 nm. The spectrum of meloxicam is presented in Figure 2; it exhibits a maximum at 362 nm. Both maxima wavelengths coincide with those of the solutions of respective drugs in sodium hydroxide.

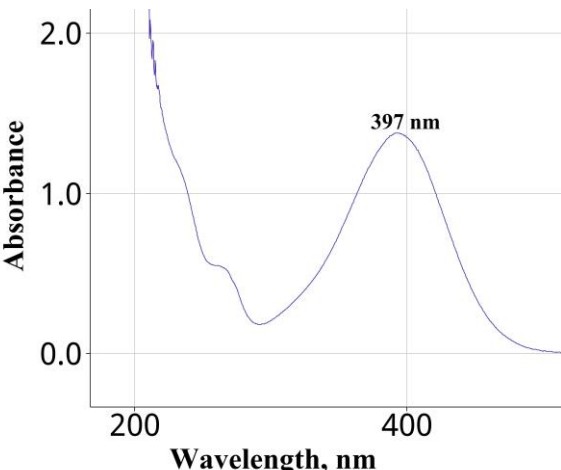

**Figure 1.** The absorption spectrum of 25 mg/L solution of nimesulide against 10% solution of sodium carbonate.

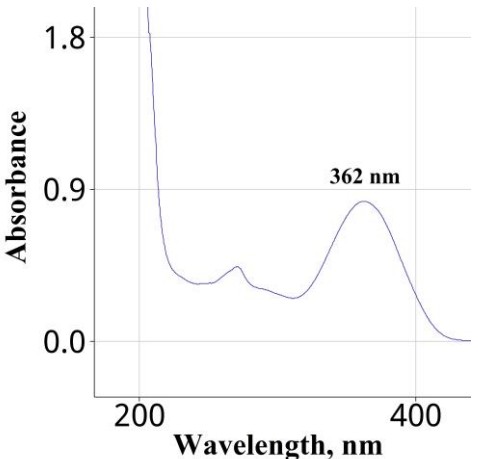

**Figure 2.** The absorption spectrum of 20 mg/L solution of meloxicam against 10% solution of sodium carbonate.

### 3.2. Selection of Sodium Carbonate Solution Concentration

The working solutions of nimesulide with a concentration of 25 mg/L and the working solution of meloxicam with a concentration of of 20 mg/L using the sodium carbonate solution with different concentrations (1, 2, 5, 10, 15 and 20%) as the solvent were prepared and their absorbances at respective wavelengths against respective solvents were measured. The results are presented in Figure 3. According to the data, the 10% sodium carbonate solution was selected as the solvent for all future experiments.

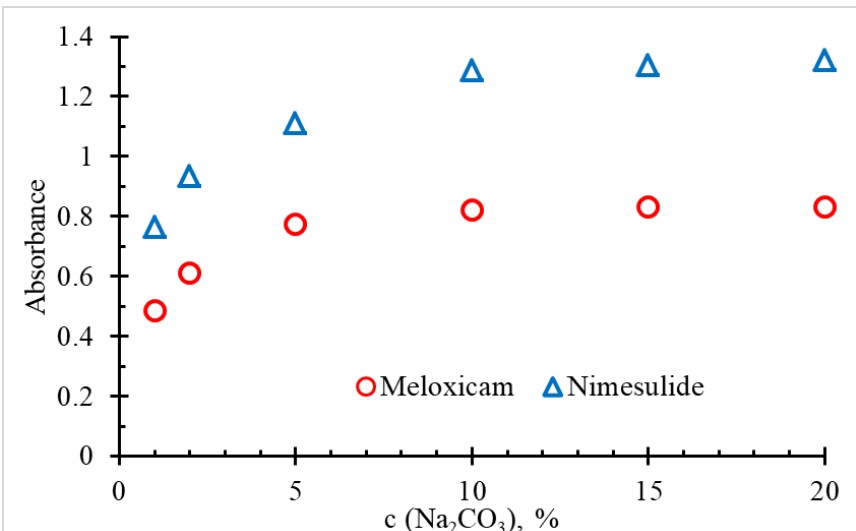

**Figure 3.** Dependence of the absorbances of nimesulide and meloxicam on the solvent concentration.

### 3.3. Construction of the Calibration Graph

The working solutions of nimesulide and meloxicam with different concentrations ranging from 5 to 25 mg/L were prepared. The absorbances of prepared solutions were measured against the 10% solution of sodium carbonate at the corresponding wavelengths. The results are presented in Figure 4.

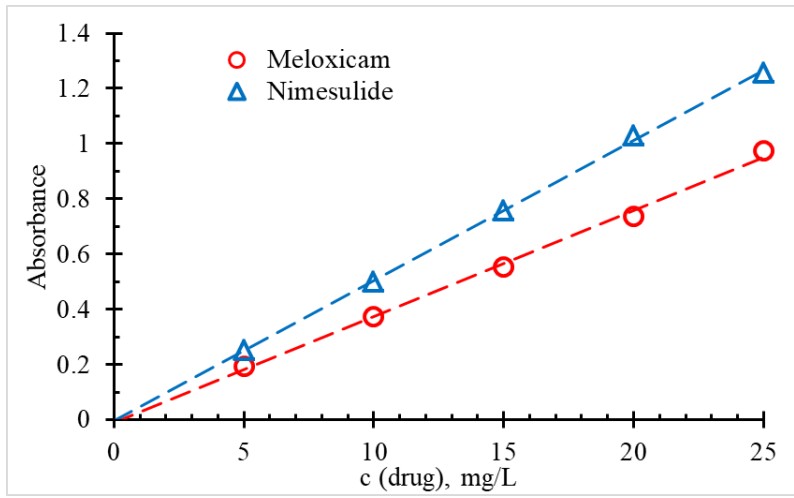

**Figure 4.** The calibration graphs for nimesulide and meloxicam.

### 3.4. Analytical Performance

The analytical performance of the method was determined in accordance with the ICH guidelines on the validation of analytical procedures [64]. The method was tested for linearity, limits of detection and quantification, selectivity, accuracy and inter- and intra-day precision.

### 3.5. Linearity

According to Figure 4, the dependences of the absorbances of the drug solutions at the corresponding wavelengths on the drug concentration were linear in the range from 5 to 25 mg/L. The regression analysis was performed using the least-squares technique [65]. Additionally, the Ringbom's optimum range [66–68], the molar attenuation coefficient and the Sandell's sensitivity coefficient [69] were calculated. The parameters of the regression equation are listed in Table 3.

**Table 3.** The parameters of the linear regression of the dependences of the absorbances of the solutions of nimesulide at 397 nm and meloxicam at 362 nm on the drug concentrations, and the analytical parameters of the methods.

| Parameter | Value | |
|---|---|---|
| Analyzed pharmaceutical ingredient | Nimesulide | Meloxicam |
| Wavelength of maximum absorbance (nm) | 397 | 362 |
| Slope and its confidence interval (f = 4, *p* = 95%) (L/mg) | $0.051 \pm 0.001$ | $0.038 \pm 0.001$ |
| Intercept and its confidence interval (f = 4, *p* = 95%) | $-0.002 \pm 0.001$ | $-0.01 \pm 0.01$ |
| $R^2$ value | 0.999 | 0.996 |
| Linearity range (mg/L) | 5–25 | 5–25 |
| Ringbom's optimum range (mg/L) | 4–14 | 6–18 |
| Molar attenuation coefficient and its confidence interval (f = 4, *p* = 95%) ($m^2$/mol) | $6100 \pm 100$ | $9100 \pm 300$ |
| Sandell's sensitivity coefficient and its confidence interval (f = 4, *p* = 95%) ($\mu g/cm^2$) | $0.019 \pm 0.002$ | $0.026 \pm 0.004$ |
| Limit of detection (mg/L) | 0.8 | 1.9 |
| Limit of quantification (mg/L) | 2.5 | 5.8 |

### 3.6. Limit of Detection and Limit of Quantification

The limit of detection and the limit of quantification of the method [64,70–72] were calculated based on the standard deviation of a linear response and a slope. The values are presented in Table 3.

### 3.7. Selectivity with Respect to Common Excipients

According to the Russian State Register of Pharmaceutical Products, tablets of nimesulide contain lactose monohydrate, sodium starch glycolate, polyvinylpyrrolidone K-17, magnesium stearate, microcrystalline cellulose and colloidal silicon dioxide as the common excipients. Tablets of meloxicam contain lactose monohydrate, talcum, magnesium stearate and microcrystalline cellulose as the common excipients. The possible interference of these excipients was studied. For that, the 1 g/L water solutions of polyvinylpyrrolidone, lactose monohydrate and sodium starch glycolate and the 1 g/L suspensions of magnesium stearate, microcrystalline cellulose and colloidal silicon dioxide in 10% solutions of sodium carbonate were prepared. The solutions were left for 60 min and their absorbances at 362 and 397 nm against the sodium carbonate solution were measured. No development of the yellow color was observed and the absorbances were less than 0.002; this indicated that the tested excipients did not interfere.

### 3.8. Accuracy

For each active pharmaceutical ingredient, ten series of experiments were conducted. For nimesulide, in the first five series, ten working solutions with each of the concentrations equal to 5, 10, 15, 20 and 25 mg/L and, in the next five series, ten sample solutions from tablets with each of the concentration equal to 5, 10, 15, 20 and 25 mg/L were prepared. The same ten series of solutions for meloxicam were prepared. The absorbances of the solutions were recorded as described in the general procedure, the concentrations of the solutions were calculated according to the regression equations and the relative uncertainties were determined. The results are collected in Table 4.

**Table 4.** The accuracy tests of the methods and for the model swab extract solutions.

| Tested Solutions of Nimesulide | Mean Measured Concentration of Nimesulide (mg/L) | Relative Uncertainty (%) | Tested Solutions of Meloxicam | Mean Measured Concentration of Meloxicam (mg/L) | Relative Uncertainty (%) |
|---|---|---|---|---|---|
| Working solution, 5 mg/L | 5.06 | 1.2 | Working solution, 5 mg/L | 5.04 | 0.8 |
| Working solution, 10 mg/L | 10.05 | 0.5 | Working solution, 10 mg/L | 10.04 | 0.4 |
| Working solution, 15 mg/L | 15.06 | 0.4 | Working solution, 15 mg/L | 15.07 | 0.5 |
| Working solution, 20 mg/L | 20.07 | 0.4 | Working solution, 20 mg/L | 20.05 | 0.3 |
| Working solution, 25 mg/L | 25.09 | 0.4 | Working solution, 25 mg/L | 25.11 | 0.4 |
| Sample solution from tablets, 5 mg/L | 4.96 | 0.8 | Sample solution from tablets, 5 mg/L | 4.94 | 1.2 |
| Sample solution from tablets, 10 mg/L | 9.95 | 0.5 | Sample solution from tablets, 10 mg/L | 9.93 | 0.7 |
| Sample solution from tablets, 15 mg/L | 14.93 | 0.5 | Sample solution from tablets, 15 mg/L | 14.95 | 0.3 |
| Sample solution from tablets, 20 mg/L | 19.95 | 0.3 | Sample solution from tablets, 20 mg/L | 19.93 | 0.4 |
| Sample solution from tablets, 25 mg/L | 24.92 | 0.3 | Sample solution from tablets, 25 mg/L | 24.91 | 0.4 |
| Swab extract from working solution, 5 mg/L | 4.79 | 4.2 | Swab extract from working solution, 5 mg/L | 4.85 | 3.0 |
| Swab extract from working solution, 10 mg/L | 9.81 | 1.9 | Swab extract from working solution, 10 mg/L | 9.87 | 1.3 |
| Swab extract from working solution, 15 mg/L | 14.76 | 1.6 | Swab extract from working solution, 15 mg/L | 14.84 | 1.1 |
| Swab extract from working solution, 20 mg/L | 19.69 | 1.2 | Swab extract from working solution, 20 mg/L | 19.73 | 1.2 |
| Swab extract from working solution, 25 mg/L | 24.72 | 1.1 | Swab extract from working solution, 25 mg/L | 24.77 | 1.0 |
| Swab extract from sample solution from tablets, 5 mg/L | 4.68 | 6.4 | Swab extract from sample solution from tablets, 5 mg/L | 4.73 | 5.4 |
| Swab extract from sample solution from tablets, 10 mg/L | 9.70 | 3.0 | Swab extract from sample solution from tablets, 10 mg/L | 9.76 | 2.4 |

| Tested Solutions of Nimesulide | Mean Measured Concentration of Nimesulide (mg/L) | Relative Uncertainty (%) | Tested Solutions of Meloxicam | Mean Measured Concentration of Meloxicam (mg/L) | Relative Uncertainty (%) |
|---|---|---|---|---|---|
| Swab extract from sample solution from tablets, 15 mg/L | 14.66 | 2.3 | Swab extract from sample solution from tablets, 15 mg/L | 14.69 | 2.0 |
| Swab extract from sample solution from tablets, 20 mg/L | 19.62 | 1.9 | Swab extract from sample solution from tablets, 20 mg/L | 19.65 | 1.7 |
| Swab extract from sample solution from tablets, 25 mg/L | 24.57 | 1.7 | Swab extract from sample solution from tablets, 25 mg/L | 24.66 | 1.4 |

*3.9. Intra-Day Precision*

For each active pharmaceutical ingredient, ten series of experiments were conducted. For nimesulide, in the first five series, ten working solutions with each of the concentrations equal to 5, 10, 15, 20 and 25 mg/L and, in the next five series, ten sample solutions from tablets with each of the concentration equal to 5, 10, 15, 20 and 25 mg/L were prepared. The same ten series of solutions for meloxicam were prepared. The absorbances of the solutions were recorded as described in the general procedure, the concentrations of the solutions were calculated according to the regression equations and the relative standard deviations were determined. The results are collected in Table 5.

**Table 5.** The precision test of the method and for the model swab extract solutions.

| Tested Solutions of Nimesulide | Standard Deviation (mg/L) | Relative Standard Deviation (%) | Tested Solutions of Meloxicam | Standard Deviation (mg/L) | Relative Standard Deviation (%) |
|---|---|---|---|---|---|
| Working solution, 5 mg/L (intra-day) | 0.07 | 1.4 | Working solution, 5 mg/L (intra-day) | 0.08 | 1.6 |
| Working solution, 10 mg/L (intra-day) | 0.11 | 1.1 | Working solution, 10 mg/L (intra-day) | 0.13 | 1.3 |
| Working solution, 15 mg/L (intra-day) | 0.14 | 0.9 | Working solution, 15 mg/L (intra-day) | 0.15 | 1.0 |
| Working solution, 20 mg/L (intra-day) | 0.14 | 0.7 | Working solution, 20 mg/L (intra-day) | 0.18 | 0.9 |
| Working solution, 25 mg/L (intra-day) | 0.15 | 0.6 | Working solution, 25 mg/L (intra-day) | 0.20 | 0.8 |
| Sample solution from tablets, 5 mg/L (intra-day) | 0.06 | 1.3 | Sample solution from tablets, 5 mg/L (intra-day) | 0.08 | 1.6 |
| Sample solution from tablets, 10 mg/L (intra-day) | 0.10 | 1.0 | Sample solution from tablets, 10 mg/L (intra-day) | 0.12 | 1.2 |
| Sample solution from tablets, 15 mg/L (intra-day) | 0.12 | 0.8 | Sample solution from tablets, 15 mg/L (intra-day) | 0.13 | 0.9 |
| Sample solution from tablets, 20 mg/L (intra-day) | 0.12 | 0.6 | Sample solution from tablets, 20 mg/L (intra-day) | 0.16 | 0.8 |

| Tested Solutions of Nimesulide | Standard Deviation (mg/L) | Relative Standard Deviation (%) | Tested Solutions of Meloxicam | Standard Deviation (mg/L) | Relative Standard Deviation (%) |
|---|---|---|---|---|---|
| Sample solution from tablets, 25 mg/L (intra-day) | 0.17 | 0.7 | Sample solution from tablets, 25 mg/L (intra-day) | 0.17 | 0.7 |
| Working solution, 5 mg/L (inter-day) | 0.09 | 1.8 | Working solution, 5 mg/L (inter-day) | 0.11 | 2.1 |
| Working solution, 10 mg/L (inter-day) | 0.12 | 1.2 | Working solution, 10 mg/L (inter-day) | 0.15 | 1.5 |
| Working solution, 15 mg/L (inter-day) | 0.15 | 1.0 | Working solution, 15 mg/L (inter-day) | 0.18 | 1.2 |
| Working solution, 20 mg/L (inter-day) | 0.16 | 0.8 | Working solution, 20 mg/L (inter-day) | 0.20 | 1.0 |
| Working solution, 25 mg/L (inter-day) | 0.18 | 0.7 | Working solution, 25 mg/L (inter-day) | 0.23 | 0.9 |
| Sample solution from tablets, 5 mg/L (inter-day) | 0.08 | 1.7 | Sample solution from tablets, 5 mg/L (inter-day) | 0.09 | 1.9 |
| Sample solution from tablets, 10 mg/L (inter-day) | 0.15 | 1.5 | Sample solution from tablets, 10 mg/L (inter-day) | 0.17 | 1.7 |
| Sample solution from tablets, 15 mg/L (inter-day) | 0.16 | 1.1 | Sample solution from tablets, 15 mg/L (inter-day) | 0.19 | 1.3 |
| Sample solution from tablets, 20 mg/L (inter-day) | 0.18 | 0.9 | Sample solution from tablets, 20 mg/L (inter-day) | 0.22 | 1.1 |
| Sample solution from tablets, 25 mg/L (inter-day) | 0.20 | 0.8 | Sample solution from tablets, 25 mg/L (inter-day) | 0.25 | 1.0 |
| Swab extract from working solution, 5 mg/L | 0.15 | 3.2 | Swab extract from working solution, 5 mg/L | 0.17 | 3.6 |
| Swab extract from working solution, 10 mg/L | 0.23 | 2.3 | Swab extract from working solution, 10 mg/L | 0.27 | 2.7 |
| Swab extract from working solution, 15 mg/L | 0.24 | 1.6 | Swab extract from working solution, 15 mg/L | 0.27 | 1.8 |
| Swab extract from working solution, 20 mg/L | 0.26 | 1.3 | Swab extract from working solution, 20 mg/L | 0.32 | 1.6 |
| Swab extract from working solution, 25 mg/L | 0.30 | 1.2 | Swab extract from working solution, 25 mg/L | 0.32 | 1.3 |
| Swab extract from sample solution from tablets, 5 mg/L | 0.16 | 3.4 | Swab extract from sample solution from tablets, 5 mg/L | 0.18 | 3.7 |
| Swab extract from sample solution from tablets, 10 mg/L | 0.26 | 2.7 | Swab extract from sample solution from tablets, 10 mg/L | 0.29 | 3.0 |

**Table 5.** *Cont.*

| Tested Solutions of Nimesulide | Standard Deviation (mg/L) | Relative Standard Deviation (%) | Tested Solutions of Meloxicam | Standard Deviation (mg/L) | Relative Standard Deviation (%) |
|---|---|---|---|---|---|
| Swab extract from sample solution from tablets, 15 mg/L | 0.28 | 1.9 | Swab extract from sample solution from tablets, 15 mg/L | 0.35 | 2.4 |
| Swab extract from sample solution from tablets, 20 mg/L | 0.31 | 1.6 | Swab extract from sample solution from tablets, 20 mg/L | 0.37 | 1.9 |
| Swab extract from sample solution from tablets, 25 mg/L | 0.34 | 1.4 | Swab extract from sample solution from tablets, 25 mg/L | 0.42 | 1.7 |

*3.10. Inter-Day Precision*

The twenty series of solutions were prepared as described in the previous section over five consecutive days. The absorbances of the solutions were recorded as described in the general procedure, the concentrations of the solutions were calculated according to the regression equations and the relative standard deviations were determined. The results are collected in Table 5.

*3.11. Accuracy for the Determination of Model Swab Extract Solutions*

For each active pharmaceutical ingredient, ten series of experiments were conducted. For nimesulide, in the first five series, ten swab extracts from working solutions with each of the concentrations equal to 5, 10, 15, 20 and 25 mg/L and, in the next five series, ten swab extracts from sample solutions from tablets with each of the concentration equal to 5, 10, 15, 20 and 25 mg/L were prepared. The same ten series of solutions for meloxicam were prepared. The absorbances of the solutions were recorded as described in the general procedure, the concentrations of the solutions were calculated according to the regression equations and the relative uncertainties were determined. The results are collected in Table 4.

*3.12. Precision for the Determination of Model Swab Extract Solutions*

For each active pharmaceutical ingredient ten series of experiments were conducted. For nimesulide, in the first five series, ten swab extracts from working solutions with each of the concentrations equal to 5, 10, 15, 20 and 25 mg/L and, in the next five series, ten swab extracts from sample solutions from tablets with each of the concentration equal to 5, 10, 15, 20 and 25 mg/L were prepared. The same ten series of solutions for meloxicam were prepared. The absorbances of the solutions were recorded as described in the general procedure, the concentrations of the solutions were calculated according to the regression equations and the relative standard deviations were determined. The results are collected in Table 5.

*3.13. Stability of Nimesulide and Meloxicam in the Sodium Carbonate Solutions*

The working solutions of nimesulide with concentration 25 mg/L and the working solution of meloxicam with concentration of 20 mg/L were prepared and left to stand at the room temperature for a week; their absorbances at respective wavelengths against the 10% solution of sodium carbonate were measured repeatedly. The results are presented in Figure 5. According to the data, the absorbance loss after one day of incubation did not exceed 1% and, after 7 days of incubation, did not exceed 5%, which means that both nimesulide and meloxicam were stable enough in the 10% sodium carbonate solution for the concentration measurement within a working day.

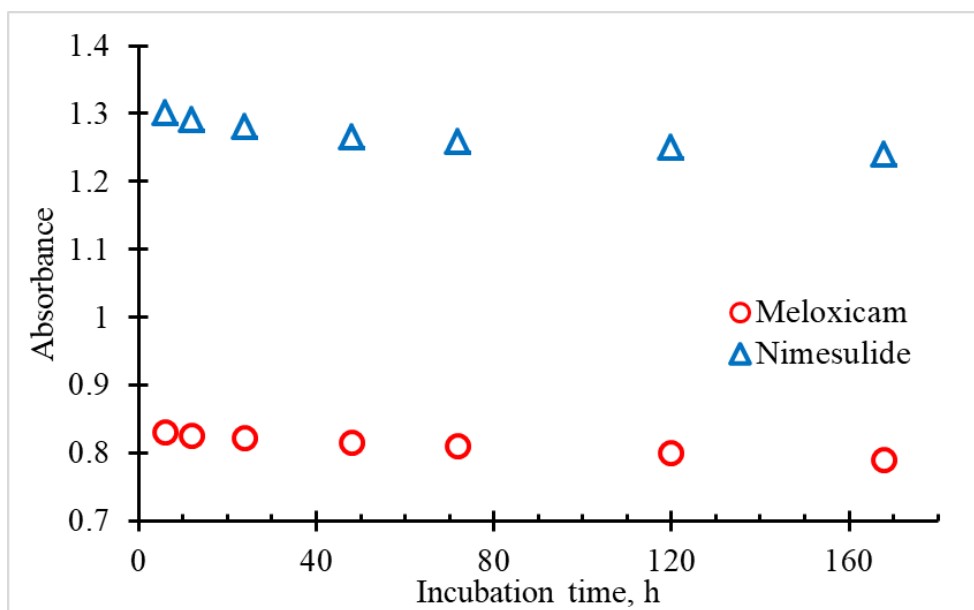

**Figure 5.** Dependence of the absorbances of nimesulide and meloxicam on the incubation time.

## 4. Discussion

The experiments showed that the proposed spectrophotometric methods are suitable for the determination of nimesulide and meloxicam in industrial equipment cleaning validation samples. The methods were rapid and simple; they did not require complicated sample preparation or sophisticated equipment. The methods were selective with respect to the common excipients, sensitive (the molar attenuation coefficient equaled 6100 m$^2$/mol for nimesulide and 9100 m$^2$/mol for meloxicam, the limit of detection equaled 0.8 mg/L for nimesulide and 1.9 mg/L for meloxicam and the limit of quantification equaled 2.5 mg/L for nimesulide and 5.8 mg/L for meloxicam), accurate (the relative uncertainty for the analysis of pharmaceutical formulations did not exceed 2%, the relative uncertainty for the analysis of the modeling swab extract did not exceed 7%, which was acceptable for cleaning validation sample analysis) and precise (the relative standard deviation did not exceed 2% for intra-, 3% for inter-day precision and 4% for the analysis of modeling swab extracts). The calibration graphs were linear, in the range from 5 to 25 mg/L of both nimesulide and meloxicam, with a good correlation coefficient. The methods are recommended for the routine and quick analysis of nimesulide and meloxicam in industrial equipment cleaning validation samples.

## 5. Conclusions

Simple spectrophotometric methods for the determination of nimesulide and meloxicam in industrial equipment cleaning validation samples using sodium carbonate were proposed. The methods were based on the colorimetric determination of basic form of the drugs in an alkaline medium. The methods showed a good analytical performance, did not require lengthy sample preparation or sophisticated laboratory equipment and were suitable for the routine analysis.

**Funding:** This research received no external funding.

**Institutional Review Board Statement:** Not applicable.

**Informed Consent Statement:** Not applicable.

**Data Availability Statement:** Not applicable.

**Conflicts of Interest:** The author was employed by LLC "Velpharm" during the period of time from February 2020 until May 2021. The paper reflects the views of the scientist and not the company.

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
