# Peer review of "UV-Spectrophotometric Determination of the Active Pharmaceutical Ingredients Meloxicam and Nimesulide in Cleaning Validation Samples with Sodium Carbonate"

_2571-8800, doi:10.3390/j6020019_

Round 1

Reviewer 1 Report (Previous Reviewer 2)

The manuscript has been adequately revised and the authors tolerably answered to my previous comments. Therefore, I recommend its acceptance for publication.

Reviewer 2 Report (Previous Reviewer 1)

The alternations made, improved the quality of the manuscript.

This manuscript is a resubmission of an earlier submission. The following is a list of the peer review reports and author responses from that submission.

Round 1

Reviewer 1 Report

The Author developed the method for UV spectrophotometric determination of nimesulide and meloxicam.

Havig read the manuscript, some questions came to my mind:

1. Are meloxicam and nimesulide stable in basic solutions such as in 10% sodium carbonate?

2. Why did Author applied the criteria from Russian Pharmacopoeia? The developed method is UV one. It is not sophisticated technique, so the general recommendation concerning validation might be applied - I suggest ICH guidelines. Russian Pharmacopoeia recommendations have only local impact.

3. How were limit of detection and limit of quantitation calculated?

4. Why did Author analyzed only one concentration for determination of accuracy and precision? It was 15 mg/L for working solution, and swab extract, and 10 mg/L for sample solution from tablets for nimesulide. For meloxicam it was 10 mg/L for working solutions, and swab extract, and 15 mg/L for sample solutions from tablets.

General conclusion: the method was not validated properly.

Reviewer 2 Report

Reviewer report on manuscript J-2203579

The submitted work aimed on the development of a UV spectrophotometric method for the determination of meloxicam and nimesulide in cleaning validation samples with sodium carbonate.

I read the manuscript carefully. The novelty of the proposed method is very poor to justify publication. Everything is standard without any exceptional scientific interest. The proposed method could be a laboratory report not a scientific paper.

 Based on the above statements, I recommend its rejection.